# Anaemia among adolescent girls, pregnant and lactating women in the southern rural region of Bangladesh: Prevalence and risk factors

Gulshan Ara[1,2]*, Rafid Hassan[1], Md. Ahshanul Haque[1], Anika Bushra Boitchi[3], Samira Dilruba Ali[1], Kazi Sudipta Kabir[1], Riad Imam Mahmud[4], Kazal Ahidul Islam[4], Hafizur Rahman[4], Zhahirul Islam[5]

1 Nutrition Research Division, icddr,b, Dhaka, Bangladesh, 2 Department of Nutrition, Sports and Exercise (NEXS), University of Copenhagen, Copenhagen, Denmark, 3 Department of Public Health and Informatics, Jahangirnagar University, Dhaka, Bangladesh, 4 Max Foundation, Dhaka, Bangladesh, 5 Infectious Disease Division, icddr,b, Dhaka, Bangladesh

* gulshan.ara@icddrb.org

**Data Availability Statement:** All relevant data are within the paper and its Supporting Information files.

## Abstract

Anaemia is a major public health concern in developing countries, particularly among children, adolescents, and women of reproductive age. The study aimed to assess the anaemia status among adolescent girls, pregnant, and lactating women with their contributing factors in the southern rural regions of Bangladesh. This cross-sectional study was conducted among 400 adolescent girls, 375 pregnant, and 375 lactating women using a multistage cluster-random sampling technique. Anaemia was measured through haemoglobin concentration in blood capillaries collected with a Hemocue 301 machine. Multinomial logistic regression was used to determine the factors associated with anaemia. The average age of pregnant and lactating women was 24 years and 15.2 years for girls. Overall, the prevalence of anaemia was 50% among pregnant women, 46% among lactating women, and 38% among adolescent girls. The risk of anaemia among adolescent girls was higher among non-Muslim (aOR = 2.13, 95%CI:1.05–4.31), belonged to families having >5 members (aOR = 2.24, 95%CI:1.16–4.31) while exposure to media reduced their risk (aOR = 0.33, 95%CI:0.15–0.74). Pregnant women who consumed a diversified diet, washed their hands after toilet, and received ≥4 ANC visits had a lower likelihood of developing anaemia. Lactating women who were employed, consumed a diversified diet, washed their hands before preparing food, and after toilet, had been exposed to media, received ≥4 ANC visits, and consumed ≥90 IFA, had a lower risk of developing anaemia. However, anaemia was more likely to be associated with lactating women who were non-Muslim (aOR = 3.75; 95% CI:1.26–11.22). The high prevalence of anaemia emphasizes the need to reconsider the existing strategy for the prevention and control of micronutrient deficiencies in Bangladesh.

**Funding:** Gulshan Ara received funding from Max Foundation, Bangladesh to conduct this study. The grant number was GR-02137. URL of funder website: https://maxfoundation.org/country/bangladesh/ The funders did not play any role in the study design, data collection and analysis, decision to publish, or preparation of the manuscript.

**Competing interests:** The authors have declared that no competing interests exist.

## Introduction

Anaemia is a global public health concern affecting almost one-third of the world's population [1]. This adverse health condition arises when the red blood cells count or the concentration of haemoglobin within them falls below normal [2]. It can appear at any stage of the human life cycle, impairing the oxygen supply to tissues. However, the increased physiological demand for iron during growth, menstruation, and pregnancy of adolescent girls and reproductive-aged women makes them more vulnerable to anaemia [3–6]. The global prevalence of anaemia is 38% and 29% among pregnant women and reproductive-aged women, respectively. Although the prevalence is heterogeneously distributed across the world, low- and middle-income countries (LMICs) bear a greater burden of anaemia [7].

A multifaceted etiology encompasses the development of anaemia. However, iron deficiency anaemia accounts for around half of all anaemic cases among women across the globe. Inadequate consumption or absorption of iron-rich foods, increased need for iron during the course of growth and pregnancy, menstrual iron loss, and intestinal parasite infections contribute to iron deficiency [5]. In addition, deficiencies in certain nutrients, including vitamin B12, folate, riboflavin, vitamin A, vitamin C, and copper, as well as protein-energy malnutrition, may lead to anaemia [2,3,6]. Anaemia can also be caused by a number of hereditary diseases including thalassemia, sickle cell anaemia, chronic inflammation, a lack of glucose-6-phosphate-dehydrogenase, and ovalocytosis [2]. Infectious diseases such as intestinal worm infestation, malaria, tuberculosis, AIDS, schistosomiasis, and tropical sprue can also contribute to anaemia [2,3]. Nevertheless, a wide range of aspects encompassing social, economic, demographic, political, ecological, biological, anthropometric, and lifestyle factors play an important role in developing anaemia [1,3,4,7–9].

Anaemia has a detrimental impact on maternal and child health. It impairs cognitive development and motor function during the early stages of growth and development [3]. Individuals suffering from anaemia experience fatigue and loss of productivity, which eventually yields poor work performance, producing a substantial economic burden to the family and nation [2,3]. Furthermore, severe anaemia increases the risk of morbidity and mortality, especially among pregnant and perinatal mothers, resulting in miscarriages, stillbirths, prematurity, and low birth weight babies [2]. The intergenerational transfer of poor iron pools from iron-deficient pregnant women to their children increases the risk of developing iron deficiency and anaemia in infants [3].

Anaemia is highly prevalent among children and women living in South Asia and Central and Western Africa [6]. Bangladesh is a South Asian developing country with a long history of struggling with anaemia [8]. In Bangladesh, anaemia affected around half of the adolescent girls (52%) [4], pregnant (50%) and lactating (49%) women [10]. According to the National Micronutrient Survey 2011–2012, only 4.8% of nonpregnant and nonlactating women (15–49 years of age) and 1.8% of school-going children suffered from iron deficiency anaemia, with approximately one-fifth of anaemic persons experiencing this condition [11]. Although the dietary intake of iron only met 8–18% of the recommended dietary allowances of children and women of reproductive age, the presence of high groundwater iron may have a salutary effect in lowering iron deficiency anaemia [9,11]. This lower prevalence of iron deficiency anaemia suggests other possible factors that could explain the occurrence of anaemia in Bangladesh. However, there is a scarcity of recent evidence of anaemia and its determinants among adolescent girls, pregnant, and lactating women in rural community settings of Bangladesh, especially in the southern region. This area, being climate-vulnerable, is more susceptible to micronutrient deficiencies. While there were numerous national surveys and studies on anaemia in Bangladesh, the majority of them were outdated [12–18], had methodological issues

[13], and were hospital-based [19–22]. However, only a small number of these studies examined the factors contributing to anaemia in adolescent girls [4], reproductive aged nonpregnant women [8,23,24], pregnant women [21,20,25], and lactating women, for whom there was no conclusive evidence. Therefore, the present study aimed to investigate the prevalence of anaemia and its determinants among adolescent girls, pregnant, and lactating women in the southern rural region of Bangladesh.

## Methods

### Study design and participants

Max Foundation has been implementing a healthy village campaign program in "Max Nutri-WASH" Program areas in 62 Unions of five districts- Patuakhali, Barguna, Khulna, Jessore, and Satkhira on water, sanitation, and hygiene, nutrition, adolescent and women reproductive health. This cross-sectional study was conducted from November to December 2021 among the beneficiaries who were enlisted in the "Max Nutri-WASH" program in three southern districts (Khulna, Patuakhali, and Satkhira) of Bangladesh. The participants of the study were subdivided into three groups: adolescent girls (10–19 years), pregnant, and lactating women.

### Sampling procedure and sample size

A multistage cluster random sampling technique was used to obtain a representative population sample in the three districts of "Max Nutri-WASH" program zones. During the first stage of sampling, three districts from the Max Nutri-WASH program area were randomly selected. In the second stage, out of 24 upazilas in the selected three districts, eight upazilas consisting of 23 unions were randomly chosen. A list of adolescent girls, pregnant and lactating women was prepared by a door-to-door screening of the illegible subjects. The next stage involved selecting a predetermined number of participants from each target group using a systematic sampling technique from that list. Due to inconsistent findings on prevalence and a lack of recent data, the sample size was determined by assuming a 50% prevalence of anaemia (p). We considered 95% confidence interval, 6% margin of error (d), 1.2 design effect, and 10% non-response rate. A total of 352 participants was determined as the sample size for each group using the following formula:

$$n = \frac{(Z\alpha/2)^2 p(1-p)}{d^2} \text{x design effect}$$

However, the final sample size for this study was 375 pregnant and lactating women within both categories and 400 adolescent girls.

### Data collection tools and procedures

A standard structured questionnaire following the questionnaire of the National Micronutrient Survey of Bangladesh, Bangladesh Demographic Health Survey was formulated to collect data from the study participants. It was developed in English and later translated into Bengali, keeping the meaning unchanged. Data were collected on sociodemographic, dietary, WASH, and reproductive history-related information from the participants. The research team trained the field staff on the questionnaire and relevant interview skills before collecting the data. They received comprehensive training on the Hemocue 301- haemoglobin measurement device. Soon after finishing the training, they pretested the questionnaire at a neighbouring location before starting the actual data collection. Written consent was obtained from each participant

before conducting the interviews. Participants provided information through face-to-face interviews with the field staff based on the pre-tested questionnaire.

## Haemoglobin measurement

The haemoglobin concentration in blood capillaries was measured using a Hemocue 301 machine (Hemocue AB, Angleholm, Sweden) to detect anaemia. A disposable lancet was used to puncture the fingertip of the middle, ring, or index finger after disinfection with 0.5% chlorohexidine gluconate. The exact first drop of blood was discarded. Gentle pressure was applied to extract the second drop. Once the blood drop appeared, the micro cuvette was immersed into it to allow capillary action to refill it with the blood specimen. Excess blood was wiped out from the edges of the microcuvettes with a piece of cotton. After turning on the hemocue photometer, the micro cuvette holder was drawn out. The blood-filled microcuvette was placed onto the holder, and the holder was stabilized. After 10–20 seconds, haemoglobin values appeared on the digital display.

## Variables of the study

The outcome variable of the study was anaemia status where the WHO threshold for haemoglobin concentration to be considered anaemic (Hb $<$11 g/dL for pregnant women and Hb $<$12 g/dL for nonpregnant women) [23]. Based on Hb levels, anaemia was further divided into mild, moderate, and severe categories for pregnant (10.0 to 10.9 g/dL, 7.0–9.9 g/dL, and 7.0 g/dL, respectively), and nonpregnant women (11.0–11.9 g/dL, 8.0–10.9 g/dL, and 8.0 g/dL, respectively) [23].

The independent variables were included in this study: age of the participants, religion (muslim, and non-muslim), educational status (primary, secondary, and higher), employment status (employed, and unemployed), family size ($<$5 members, and $\geq$5 members), dietary diversity ($<$5 food groups, and $\geq$5 food groups), toilet facilities (improved, and unimproved), handwashing before food preparation, and after toilet (occasional, and always), media exposure (yes, and no), region (Patuakhali Sadar, Rangabali, Galachipa, Batiaghata, Dumuria, Satkhira Sadar, Tala, and Kalaroa). In addition, pregnant and lactating women were interviewed to collect information about their pregnancy conditions including pregnancy trimesters (1st trimester, 2nd trimester, and 3rd trimester), no of antenatal care (ANC) visit (1–3 times, and $\geq$4 times), and iron and folic acid supplementation (IFA) (IFA $\leq$90, IFA $>$90) during their current and previous pregnancies. The Minimum Dietary Diversity-Women (MDD-W) indicator was used to assess their dietary diversity, which was developed based on a preceding 24-hour dietary recall from a list of ten selected food groups, including grains/roots/tubers, pulses/legumes, nuts/seeds, dairy, eggs, meat/poultry/fish, dark green leafy vegetables, other vitamin A-rich fruits and vegetables, other fruit, and other vegetables. A diet was considered diversified if anyone consumed at least five of the ten food groups [26]. Media exposure was constructed based on whether the participants watched television, or listened to the radio, or read the newspaper during the last week of the survey [10]. The wealth index was constructed based on principal component analysis (PCA) of the key socioeconomic variables and classified households into poorest, poorer, middle, richer, and richest [27].

## Ethical considerations

The institutional review board (IRB) of icddr,b approved the study (protocol # PR-21124). Prior to the interview, informed written consent was taken in the local language from the study participants ($>$18 years). Assent was taken from the adolescent girls whose age was below 18 years and subsequently, consent was obtained from their parents and caregivers.

Participants who were illiterate provided consent by thumb impression. The research team described to the participants about background and objectives of the study, the voluntary nature of participation, and the future use of data. They were given assurance that confidentiality would be maintained for all the gathered information. None other than the investigators of this research, the Ethical Review Committee of icddr,b, and any law-enforcing agency in the event of necessity would have access to the information.

## Statistical analysis

Descriptive statistics were applied to collectively describe background characteristics. The effect of each independent variable on the anaemia outcome (no anaemia, mild anaemia, and moderate or severe anaemia) was determined through bivariate and multivariate analyses. In the multivariate analysis, multinomial logistic regression models were used where the independent variables (with $p<0.25$ in the bivariate analysis and low multicollinearity) were taken into consideration [28]. The results are presented as adjusted odds ratios (AORs) along with 95% confidence intervals (CIs). $p<0.05$ were considered statistically significant. STATA 15 was used for all statistical analyses.

## Results

### Background characteristics

Table 1 presents the socioeconomic and demographic characteristics of the study participants. The mean age of the adolescent girls, pregnant, and lactating women was 15.2, 24.2, and 24.6 years, respectively. Most of the participants were Muslim, unemployed, and acquired secondary level education. Only 10% of the pregnant, and lactating women were employed. The average dietary diversity score was approximately 3.3 among the participants. Approximately 80% of all participants did not consume a diversified diet (five or more food groups). An equal proportion of adolescent girls and lactating women (approximately 63%) had improved toilet facilities, whereas only half of the pregnant women could avail themselves of the same facility in their households. In case of handwashing practice, over half of the participants always washed their hands before preparing food, and approximately 90% always washed their hands after using the toilet. Approximately 80% of pregnant women were in the second or third trimester of pregnancy. Among all pregnant women, 75% received ANC 1–3 times, and 23.4% consumed >90 IFA tablets. Almost half of lactating women received ANC check-ups 1–3 times during their last pregnancy, and 54% of them consumed >90 IFA tablets.

### Prevalence of anaemia among study participants

The prevalence of anaemia among adolescent girls, pregnant, and lactating women was approximately 38%, 50%, and 46%, respectively. Lactating women (28%) were more likely to suffer from mild anaemia than pregnant women (25%), and adolescent girls (23%). However, a higher prevalence of moderate or severe anaemia was found among pregnant women (25%) than lactating women (17%), and adolescent girls (15%) (Fig 1).

### Risk factors of anaemia among adolescent girls

The multiple multinomial logistic analyses in Table 2 show the factors associated with anaemia among adolescent girls in this study. Non-Muslim adolescent girls had 2.1 (aOR = 2.13, 95% CI: 1.05–4.31, p = 0.036) times higher odds of being mildly anaemic than those who practised Islam. Participants who lived in families having ≥5 members were 2.24 times (aOR = 2.24, 95% CI: 1.16–4.31, p = 0.016) more likely to be moderately/severely anaemic than those who

**Table 1. Background characteristics of the study participants.**

| Variables | Adolescent girls, n (%) | Pregnant women, n (%) | Lactating women, n (%) |
|---|---|---|---|
| **Age** | | | |
| Mean (SD) | 15.2 (1.7) | 24.2 (5.5) | 24.6 (5.6) |
| **Religion** | | | |
| Muslim | 299 (74.7) | 303 (80.8) | 292 (77.9) |
| Non-Muslim | 101 (25.3) | 72 (19.2) | 83 (22.1) |
| **Educational status** | | | |
| Primary | 25 (6.3) | 75 (20.0) | 74 (19.8) |
| Secondary | 336 (84.0) | 217 (57.9) | 224 (59.7) |
| Higher | 39 (9.7) | 83 (22.1) | 77 (20.5) |
| **Employment status** | | | |
| Employed | 12 (3.0) | 40 (10.7) | 34 (9.1) |
| Unemployed | 388 (97.0) | 335 (89.3) | 341(90.9) |
| **Family members** | | | |
| Mean (SD) | 4.7 (1.6) | 4.6 (2.0) | 5.6 (2.1) |
| <5 members | 215 (53.7) | 224 (59.7) | 115 (30.7) |
| ≥5 members | 185 (46.3) | 151 (40.3) | 260 (69.3) |
| **Dietary diversity** | | | |
| Mean (SD) | 3.3 (1.5) | 3.5 (1.6) | 3.3 (1.4) |
| <5 food group | 321 (80.2) | 282 (75.2) | 310 (82.7) |
| ≥5 food groups | 79 (19.8) | 93 (24.8) | 65 (17.3) |
| **Toilet facilities** | | | |
| Improved | 252 (63.0) | 186 (49.6) | 237 (63.2) |
| Unimproved | 148 (37.0) | 189 (50.4) | 138 (36.8) |
| **Washed hand before food preparation** | | | |
| Occasionally | 186 (46.5) | 173 (46.1) | 157 (41.9) |
| Always | 214 (53.5) | 202 (53.9) | 218 (58.1) |
| **Washed hand after toilet** | | | |
| Occasionally | 45 (11.3) | 45 (12.0) | 42 (11.2) |
| Always | 355 (88.7) | 330 (88.0) | 333 (88.8) |
| **Media exposure** | | | |
| Yes | 346 (86.5) | 295 (78.7) | 304 (81.1) |
| No | 54 (13.5) | 80 (21.3) | 71 (18.9) |
| **Pregnancy trimester** | | | |
| 1st trimester | - | 73 (19.4) | - |
| 2nd trimester | - | 160 (42.7) | - |
| 3rd trimester | - | 142 (37.9) | - |
| **No. of ANC visit** | | | |
| 1–3 times | - | 250 (75.8) | 211 (51.3) |
| ≥4 times | - | 80 (24.2) | 164 (43.7) |
| **No. of IFA intake** | | | |
| IFA ≤90 | - | 206 (76.6) | 173 (46.1) |
| IFA >90 | - | 63 (23.4) | 202 (53.9) |
| **Wealth index** | | | |
| Poorest | 80 (20.0) | 92 (24.5) | 72 (19.2) |
| Poorer | 85 (21.2) | 63 (16.8) | 73 (19.5) |
| Middle | 77 (19.3) | 67 (17.9) | 83 (22.1) |
| Richer | 78 (19.5) | 75 (20.0) | 78 (20.8) |

(*Continued*)

**Table 1.** (Continued)

| Variables | Adolescent girls, n (%) | Pregnant women, n (%) | Lactating women, n (%) |
|---|---|---|---|
| Richest | 80 (20.0) | 78 (20.8) | 69 (18.4) |
| **Region** | | | |
| Patuakhali Sadar | 61 (15.3) | 57 (15.2) | 58 (15.5) |
| Rangabali | 30 (7.5) | 31 (8.2) | 32 (8.5) |
| Galachipa | 53 (13.3) | 42 (11.2) | 44 (11.7) |
| Batiaghata | 23 (5.7) | 16 (4.3) | 20 (5.3) |
| Dumuria | 142 (35.5) | 118 (31.5) | 129 (34.4) |
| Satkhira Sadar | 34 (8.5) | 53 (14.1) | 17 (4.6) |
| Tala | 44 (11.0) | 43 (11.5) | 56 (14.9) |
| Kalaroa | 13 (3.2) | 15 (4.0) | 19 (5.1) |

n, number; %, percentage.

- no observation.

had smaller families (<5 members). Exposure to media reduced the risk of developing moderate or severe anaemia among adolescent girls by 67% (aOR = 0.33, 95% CI: 0.15–0.74, p = 0.006). Adolescent girls residing in Rangabali (aOR = 0.21, 95% CI: 0.06–0.79, p = 0.021), Dumuria (aOR = 0.25, 95% CI: 0.09–0.67, p = 0.006), and Tala (aOR = 0.10, 95% CI: 0.02–0.48, p = 0.004) had lower odds of being moderate or severe anaemia. Similarly, girls who lived in Rangabali (aOR = 0.31, 95% CI: 0.09–1.00, p = 0.049) and Dumuria (aOR = 0.37, 95% CI: 0.16–0.87, p = 0.023) had a lower likelihood of being mildly anaemic.

## Risk factors of anaemia among pregnant women

Table 3 presents the factors associated with anaemia among pregnant women. Women with higher educational attainment were more likely to have mild anaemia than those with primary

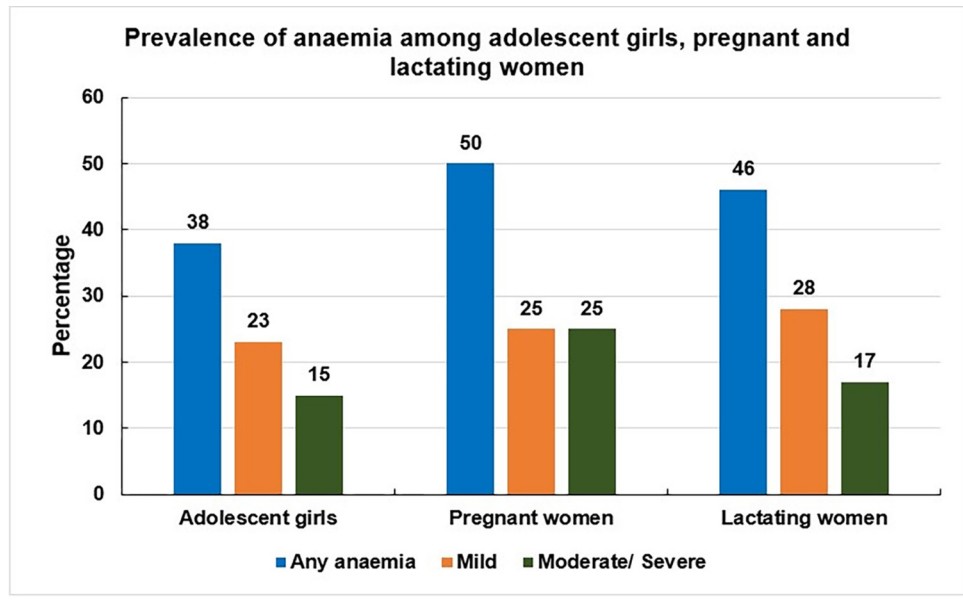

**Fig 1. Prevalence of anaemia among the study participants.**

**Table 2. Factors associated with anaemia among adolescent girls in Bangladesh (n = 400).**

| Characteristics | Mild Anaemia | | Moderate/severe Anaemia | |
|---|---|---|---|---|
| | aOR [95%CI] | p-value | aOR [95%CI] | p-value |
| **Religion** | | | | |
| Muslim | Ref | | Ref | |
| Non-Muslim | 2.13 [1.05–4.31] | 0.036 | 1.24 [0.48–3.19] | 0.653 |
| **Family members** | | | | |
| <5 members | Ref | | Ref | |
| ≥5 members | 1.09 [0.65–1.84] | 0.732 | 2.24 [1.16–4.31] | 0.016 |
| **Dietary diversity** | | | | |
| <5 food group | Ref | | Ref | |
| ≥5 food groups | 1.35 [0.73–2.50] | 0.340 | 0.65 [0.26–1.64] | 0.364 |
| **Toilet facilities** | | | | |
| Unimproved | Ref | | Ref | |
| Improved | 0.61 [0.37–1.02] | 0.062 | 0.80 [0.43–1.52] | 0.501 |
| **Washed hand before food preparation** | | | | |
| Occasionally | Ref | | Ref | |
| Always | 1.35 [0.80–2.29] | 0.260 | 0.84 [0.45–1.57] | 0.576 |
| **Media exposure** | | | | |
| No | Ref | | Ref | |
| Yes | 0.58 [0.27–1.23] | 0.157 | 0.33 [0.15–0.74] | 0.006 |
| **Wealth index** | | | | |
| Poorest | Ref | | Ref | |
| Poorer | 0.71 [0.32–1.57] | 0.397 | 0.99 [0.40–2.47] | 0.984 |
| Middle | 0.73 [0.33–1.64] | 0.447 | 0.51 [0.18–1.45] | 0.206 |
| Richer | 0.89 [0.40–1.95] | 0.767 | 0.81 [0.30–2.14] | 0.667 |
| Richest | 0.70 [0.31–1.58] | 0.386 | 0.83 [0.30–2.28] | 0.718 |
| **Region** | | | | |
| Patuakhali Sadar | Ref | | Ref | |
| Rangabali | 0.31 [0.09–1.00] | 0.049 | 0.21 [0.06–0.79] | 0.021 |
| Galachipa | 0.54 [0.21–1.37] | 0.195 | 0.38 [0.13–1.07] | 0.068 |
| Batiaghata | 0.30 [0.08–1.15] | 0.079 | 0.28 [0.06–1.34] | 0.111 |
| Dumuria | 0.37 [0.16–0.87] | 0.023 | 0.25 [0.09–0.67] | 0.006 |
| Satkhira Sadar | 0.53 [0.18–1.56] | 0.247 | 0.67 [0.22–2.11] | 0.499 |
| Tala | 0.36 [0.13–1.00] | 0.050 | 0.10 [0.02–0.48] | 0.004 |
| Kalaroa | 0.89 [0.19–4.21] | 0.878 | 1.50 [0.33–6.92] | 0.602 |

aOR: Adjusted odds ratio; Ref: Reference; CI: Confidence interval.

education (aOR = 4.72, 95% CI: 1.58–14.11; p = 0.006). Pregnant women who consumed a diversified diet (≥5 food groups) had an 86% lower likelihood of being mildly anaemic (aOR = 0.14, 95% CI: 0.05–0.37, p<0.001). The pregnant women who always washed their hands after using the toilet had a lower risk of suffering from mild (aOR = 0.23, 95% CI: 0.08–0.68, p = 0.008) and moderate or severe (aOR = 0.16, 95% CI: 0.06–0.44, p<0.001) anaemia than the women who washed their hands occasionally. Furthermore, pregnant women in their second and third trimesters had around 10 times (aOR = 9.66, 95% CI: 2.88–32.41; p<0.001) and 13 times (aOR = 13.49, 4.02–45.21; p<0.001) higher chances of developing mild anaemia, respectively, compared to those in the first trimester. However, higher odds of moderate to

**Table 3. Factors associated with anaemia among pregnant women in Bangladesh (n = 375).**

| Characteristics | Mild Anaemia | | Moderate/severe Anaemia | |
|---|---|---|---|---|
| | aOR [95%CI] | p-value | aOR [95%CI] | p-value |
| **Educational status** | | | | |
| Primary | Ref | | Ref | |
| Secondary | 1.16 [0.46–2.88] | 0.756 | 0.62 [0.27–1.44] | 0.270 |
| Higher | 4.72 [1.58–14.11] | 0.006 | 1.43 [0.47–4.31] | 0.530 |
| **Dietary diversity** | | | | |
| <5 food group | Ref | | Ref | |
| ≥5 food groups | 0.14 [0.05–0.37] | <0.001 | 0.67 [0.32–1.42] | 0.294 |
| **Toilet facilities** | | | | |
| Unimproved | Ref | | Ref | |
| Improved | 0.63 [0.32–1.25] | 0.190 | 0.65 [0.31–1.34] | 0.246 |
| **Washed hand after toilet** | | | | |
| Occasionally | Ref | | Ref | |
| Always | 0.23 [0.08–0.68] | 0.008 | 0.16 [0.06–0.44] | <0.001 |
| **Pregnancy trimester** | | | | |
| 1st trimester | | | | |
| 2nd trimester | 9.66 [2.88–32.41] | <0.001 | 9.48 [2.89–31.03] | <0.001 |
| 3rd trimester | 13.49 [4.02–45.21] | <0.001 | 10.28 [3.07–34.44] | <0.001 |
| **No. of ANC visit** | | | | |
| 1–3 times | Ref | | Ref | |
| ≥4 times | 0.35 [0.16–0.77] | 0.009 | 0.24 [0.10–0.56] | 0.001 |
| **Wealth index** | | | | |
| Poorest | Ref | | Ref | |
| Poorer | 1.73 [0.58–5.14] | 0.322 | 1.21 [0.41–3.55] | 0.728 |
| Middle | 2.42 [0.85–6.88] | 0.097 | 1.56 [0.56–4.39] | 0.398 |
| Richer | 3.61 [1.25–10.41] | 0.017 | 2.03 [0.68–6.07] | 0.203 |
| Richest | 2.83 [0.95–8.45] | 0.063 | 2.10 [0.70–6.34] | 0.188 |
| **Region** | | | | |
| Patuakhali Sadar | Ref | | Ref | |
| Rangabali | 0.70 [0.15–3.23] | 0.646 | 0.36 [0.08–1.60] | 0.180 |
| Galachipa | 0.80 [0.24–2.63] | 0.712 | 0.40 [0.11–1.41] | 0.153 |
| Batiaghata | 1.09 [0.21–5.59] | 0.919 | 1.84 [0.42–8.06] | 0.418 |
| Dumuria | 0.88 [0.33–2.39] | 0.809 | 0.69 [0.26–1.77] | 0.436 |
| Satkhira Sadar | 0.47 [0.14–1.56] | 0.216 | 0.22 [0.06–0.79] | 0.020 |
| Tala | 0.36 [0.10–1.32] | 0.124 | 0.09 [0.02–0.49] | 0.005 |
| Kalaroa | 0.98 [0.16–6.06] | 0.981 | 1.10 [0.22–5.41] | 0.908 |

aOR: Adjusted odds ratio; Ref: Reference; CI: Confidence interval.

severe anaemia were also observed among women in their second (aOR = 9.48, 95% CI: 2.89–31.03; p<0.001), and third trimesters (aOR = 10.28, 3.07–34.44; p<0.001). Additionally, individuals who received ANC at least four times had a 65% (aOR = 0.35, 95% CI: 0.16–0.77, p = 0.009), and 76% (aOR = 0.24, 95% CI: 0.10–0.56, p = 0.001) lower likelihood of experiencing mild and moderate or severe anaemia, respectively. Furthermore, pregnant women who were in the wealthiest quintile were more likely than those in the lowest quintile to have mild anaemia (aOR = 3.61, 95% CI: 1.25–10.41; p = 0.017). Pregnant women living in Satkhira Sadar (aOR = 0.22, 95% CI: 0.06–0.79, p = 0.02), and Tala (aOR = 0.09, 95% CI: 0.02–0.49,

p = 0.005) were more likely to be moderate or severe anaemic compared to those who live in Patuakhali Sadar.

### Risk factors of anaemia among lactating women

The factors related to anaemia among lactating women are shown in Table 4. Employment status was related to anaemia prevalence, with working women exhibiting reduced risk of moderate to severe anaemia (aOR = 0.06, 95% CI: 0.01–0.61; p = 0.017). Lactating women who consumed foods from five or more food groups had 72% (aOR = 0.28, 95% CI: 0.12–0.65; p = 0.003) and 92% (aOR = 0.08, 95% CI: 0.02–0.36; p = 0.001) lower odds of experiencing mild and moderate to severe anaemia, respectively. Handwashing practices also played a role, as regular hand washing before food preparation was associated with lower risk of mild anaemia (aOR = 0.37, 95% CI: 0.20–0.67; p = 0.001) than those who washed their hands occasionally. Furthermore, lactating women had lower odds of being moderately or severely anaemic if they always washed their hands after toilet usage (aOR = 0.27, 95% CI: 0.09–0.78; p = 0.016). Exposure to media decreased the likelihood of moderate to severe anaemia among lactating mothers (aOR = 0.35, 95% CI: 0.15–0.85; p = 0.020). Both mild and moderate to severe anaemia was negatively associated with ANC visits and IFA intake. Lactating women who received four or more ANC visits during their last pregnancy had 70% lower odds of mild (aOR = 0.30, 95% CI: 0.17–0.55; p<0.001) and moderate to severe anaemia (aOR = 0.28, 95% CI: 0.13–0.58; p<0.001). In terms of their IFA intake, those who consumed more than 90 tablets had a 55% (aOR = 0.45, 95% CI: 0.25–0.79; p = 0.005) lower chance of having mild anaemia and an 81% (aOR = 0.19, 95% CI: 0.09–0.39; p<0.001) lower chance of having moderate or severe anaemia. Residence in certain areas also impacted anaemia prevalence among lactating women. Those residing in Rangabali, Galachipa, Dumuria, and Kalaroa had lower odds of mild anaemia, while Rangabali and Dumuria residents had higher odds of moderate to severe anaemia. Non-Muslim lactating women had 3.8 times (aOR = 3.75, 95% CI: 1.26–11.22; p = 0.018) higher likelihood of having moderate-to-severe anaemia than Muslim women. In addition, lactating women in the middle wealth quintile had 3.3 times (aOR = 3.26, 95% CI: 1.35–7.89; p = 0.009) higher odds of being mildly anaemic.

## Discussion

Anaemia is still a major cause of public health concern in Bangladesh. The current study aimed to determine the prevalence of anaemia among adolescent girls, pregnant, and lactating women in the southern rural region of the country. Furthermore, it aimed to identify the potential risk factors associated with anaemia in these specific population groups. Anaemia was prevalent among 38% of adolescent girls, 50% of pregnant women, and 46% of lactating mothers. According to the study, the following factors were associated with anaemia: among adolescent girls, religion, family members, media exposure, and region; among pregnant women, dietary diversity, handwashing after using the toilet, pregnancy trimester, number of ANC visits, and region. Employment status, religion, dietary diversity, handwashing practices, media exposure, number of ANC visits, IFA intake, and region were found to be associated factors among lactating women.

The prevalence of anaemia among adolescent girls was nearly concordant with previous studies conducted in Bangladesh (43%) [4], and Nepal (38%) [29]. Conversely, this study found comparatively lower prevalence when compared to the prevalence rates reported in studies from Bangladesh (48–52%) [4,10], India (48–85%), Sri Lanka (58.1%) [4], Nepal (52–66%) [29]. In case of pregnant women, anaemia prevalence was almost comparable with the earlier findings, in Bangladesh (50–56%) [10,30]. However, the findings were in contrast to

**Table 4. Factors associated with anaemia among lactating women in Bangladesh (n = 375).**

| Characteristics | Mild Anaemia | | Moderate/severe Anaemia | |
|---|---|---|---|---|
| | aOR [95%CI] | p-value | aOR [95%CI] | p-value |
| **Educational status** | | | | |
| Primary | Ref | | Ref | |
| Secondary | 0.90 [0.41–1.99] | 0.803 | 0.58 [0.23–1.46] | 0.246 |
| Higher | 0.95 [0.34–2.65] | 0.926 | 0.74 [0.22–2.43] | 0.618 |
| **Employment status** | | | | |
| Unemployed | Ref | | Ref | |
| Employed | 0.64 [0.22–1.84] | 0.407 | 0.06 [0.01–0.61] | 0.017 |
| **Religion** | | | | |
| Muslim | Ref | | Ref | |
| Non-Muslim | 2.18 [0.92–5.14] | 0.076 | 3.75 [1.26–11.22] | 0.018 |
| **Family members** | | | | |
| <5 members | Ref | | Ref | |
| ≥5 members | 1.40 [0.75–2.60] | 0.287 | 0.88 [0.42–1.83] | 0.725 |
| **Dietary diversity** | | | | |
| <5 food group | Ref | | Ref | |
| ≥5 food groups | 0.28 [0.12–0.65] | 0.003 | 0.08 [0.02–.0.36] | 0.001 |
| **Toilet facilities** | | | | |
| Unimproved | Ref | | Ref | |
| Improved | 1.01 [0.56–1.82] | 0.975 | 0.72 [0.35–1.46] | 0.363 |
| **Washed hand before food preparation** | | | | |
| Occasionally | Ref | | Ref | |
| Always | 0.37 [0.20–0.67] | 0.001 | 0.78 [0.37–1.65] | 0.515 |
| **Washed hand after toilet** | | | | |
| Occasionally | Ref | | Ref | |
| Always | 0.72 [0.27–1.91] | 0.514 | 0.27 [0.09–0.78] | 0.016 |
| **Media exposure** | | | | |
| No | Ref | | Ref | |
| Yes | 0.72 [0.33–1.57] | 0.405 | 0.35 [0.15–0.85] | 0.020 |
| **No. of ANC visit** | | | | |
| 1–3 times | Ref | | Ref | |
| ≥4 times | 0.30 [0.17–0.55] | <0.001 | 0.28 [0.13–0.58] | 0.001 |
| **No. of IFA intake** | | | | |
| IFA ≤90 | Ref | | Ref | |
| IFA >90 | 0.45 [0.25–0.79] | 0.005 | 0.19 [0.09–0.39] | <0.001 |
| **Wealth index** | | | | |
| Poorest | Ref | | Ref | |
| Poorer | 1.24 [0.51–3.05] | 0.634 | 0.64 [0.21–1.94] | 0.427 |
| Middle | 3.26 [1.35–7.89] | 0.009 | 2.05 [0.70–6.05] | 0.192 |
| Richer | 1.83 [0.70–4.79] | 0.221 | 2.87 [0.93–8.89] | 0.067 |
| Richest | 1.61 [0.58–4.52] | 0.363 | 2.72 [0.79–9.37] | 0.113 |
| **Region** | | | | |
| Patuakhali Sadar | Ref | | Ref | |
| Rangabali | 0.14 [0.04–0.50] | 0.003 | 0.19 [0.05–0.81] | 0.024 |
| Galachipa | 0.24 [0.08–0.71] | 0.01 | 0.33 [0.09–1.16] | 0.083 |
| Batiaghata | 2.51 [0.49–12.78] | 0.268 | 2.31 [0.28–18.79] | 0.435 |
| Dumuria | 0.24 [0.09–0.61] | 0.003 | 0.16 [0.05–0.58] | 0.005 |

*(Continued)*

**Table 4.** (Continued)

| Characteristics | Mild Anaemia | | Moderate/severe Anaemia | |
|---|---|---|---|---|
| | aOR [95%CI] | p-value | aOR [95%CI] | p-value |
| Satkhira Sadar | 0.29 [0.07–1.21] | 0.089 | 0.36 [0.05–2.52] | 0.306 |
| Tala | 0.45 [0.16–1.25] | 0.126 | 0.69 [0.20–2.44] | 0.568 |
| Kalaroa | 0.16 [0.03–0.82] | 0.028 | 0.39 [0.07–2.18] | 0.282 |

aOR: Adjusted odds ratio; Ref: Reference; CI: Confidence interval.

several studies conducted in Bangladesh (37%, 40%, 59%, 63%) [21], Ethiopia (23%) [31], Bhutan, Sri Lanka, Nepal (59–65%) [22], India (87%) [32]. Similarly, high prevalence of anaemia among lactating women was noticed in many studies conducted in Bangladesh (48%) [10], Sierra Leone (53%) [33], Mozambique (53%), Tanzania (46%), Kenia (44%) [3]. Conversely, a lower prevalence also observed in East Africa (36%), Rwanda (19%) [34], and Ethiopia (22%) [28]. Regarding the levels of anaemia status, moderate to severe anaemia was more prevalent among pregnant, and lactating women than adolescent girls. This finding was comparable with the recent Bangladesh Demographic Health Survey [10]. A difference in the prevalence of anaemia between and within countries has been shown. However, the variation might be due to differences in sociodemographic and cultural issues, dietary patterns, study design and methodology, and time differences of the studies [7,35].

Sociodemographic characteristics play a significant role in determining anaemia. Likewise, in other studies, lactating mothers who were employed had lower odds of having anaemia [28,34]. This might be due to their better decision-making autonomy resulting from their financial contributions to family, which empowered them to purchase a variety of foods and seek health facilities [28]. Besides, adolescent girls who had a family size of five and more were more likely to have moderate/severe anaemia, which was consistent with earlier studies [36,37]. The inability to afford micronutrient-rich foods in large families, coupled with low care, could be the possible cause [36]. This study also observed reduced likelihood of moderate to severe anaemia among adolescent girls and lactating women who had exposure to mass media. The importance of mass media was also acknowledged in earlier studies in reducing the risk of anaemia [38,39]. Mass media is considered a good source of receiving wider information regarding health-related programs which helps to increase nutrition knowledge and proper utilization of health care services and encourages women to intake diversified nutritious meals, and iron and folic acid supplements [39,40]. In our study, adolescent girls and lactating women from non-Muslim families had a considerably higher likelihood of developing anaemia than Muslims. This result was consistent with other studies conducted in Bangladesh [8], and India [32]. This difference in anaemia across religions may be due to different dietary practices and food taboos [32]. The bioavailability of dietary iron might fluctuate within particular religious communities due to distinct belief regarding food consumption, which limit their consumption of iron-rich foods [41]. The majority of non-Muslims in Bangladesh are Hindus, and most of them follow a vegetarian diet. In contrast, Muslims in this country consume more foods from animal sources than non-Muslims [8].

Inappropriate dietary habits may contribute to the development of anaemia, where low dietary diversity plays a significant role. In our study, anaemia among pregnant and lactating women was linked to the consumption of an inadequate diversified diet, which was consistent with earlier studies [31,42]. The physiological state of highly demanding nutrients during pregnancy and lactation suggests that women should consume a diversified diet to achieve nutrient adequacy to prevent the development of micronutrient-deficient anaemia [31].

Water, sanitation, and hygiene (WASH) practices are considered one of the interventions for the prevention and control of anaemia [7]. Particularly in Bangladesh, due to the tropical temperature, improper hand washing practices after excretion, before preparing meals or consuming food, and using unsanitary latrines could result in gastrointestinal parasitic infestation [8,39,43–45]. Poor general health and/or chronic blood loss through gastrointestinal parasite infestation can be considered for the association of the unhygienic practice of using toilets with developing anaemia [8]. An earlier study revealed a higher prevalence of anaemia among adolescent girls, with approximately 32% of anaemic individuals found to be infected with intestinal parasites, compared to 30.6% of non-anemic cases [44]. A multi-country study among women of reproductive age in Bangladesh, Maldives, and Nepal found that access to safe drinking water seemed to be protective factor for lowering the cases of anaemia among women in Nepal, and Bangladesh [23]. In the current study, we found that frequent handwashing after using toilets and before food preparation decreased the likelihood of developing anaemia among pregnant and lactating women.

Align with earlier studies, our study found that pregnant and lactating women who received ANC ≥4 times during their pregnancy had a lower risk of being anaemic [42,46]. This may be due to a routine diagnosis of anaemia during the ANC visit and scheduled counselling on nutrition-related knowledge, supplementation and treatment [42]. We found a negative association between IFA intake and anaemia among lactating women during their preceding pregnancy. A study conducted in India found that IFA supplementation was similarly inversely related to the incidence of anaemia, with regular use of IFA or folic acid supplements in the preceding trimester leading to a 74% decreased risk of anaemia [47]. WHO strongly recommends daily supplementation of oral iron and folic acid to reduce the risk of anaemia during pregnancy and prevent the occurrence of low birth weight [2]. During pregnancy, anaemia can act as a risk factor and impose life-threatening consequences on the mother as well as the foetus [48]. This is likely because of the depletion of iron reserves as pregnancy develops, which further increases the requirement for iron, especially during the second and third trimesters [49]. During pregnancy, particularly after the first trimester, there is a need for the maternal blood volume to expand to support foetal growth. A previous study found that the prevalence of moderate to severe anaemia increased significantly after five months of gestation and that the risk was amplified in the later stages of pregnancy [38]. In the present study, women in the second and third pregnancy trimesters were more likely to be anaemic, which aligns with earlier studies conducted in Bangladesh [22], Cameroon [44], and Ethiopia [31].

However, the study has some limitations, including the inability to diagnose iron deficiency anaemia due to the lack of data on absolute dietary iron consumption or serum ferritin levels. Being cross-sectional in nature, the data should be utilized with caution when interpreting the temporal association between sociodemographic and anaemia. Additionally, we did not collect information on further health issues, including a family history of thalassemia, a malaria infection, an obstetrical issue, helminth infections, etc. Our understanding would be improved by supplementary research on these topics; therefore, extensive investigation into the causes of anaemia is required to identify the most appropriate measures. Attention must be paid to the high prevalence of anaemia in adolescent girls, pregnant and lactating women. To improve the status of anaemia in these vulnerable populations in the southern area of Bangladesh, dietary diversity, supplementation with multiple micronutrients, and social behaviour change communication should be advocated.

## Conclusion

Our study acknowledges the importance of addressing anaemia as a major health concern among adolescent girls, pregnant and lactating women in Bangladesh. Socioeconomic

conditions dietary diversity, pregnancy status, antenatal care, IFA intake, and WASH practices contribute to the development of anaemia. Even though the national strategy to control and prevent anaemia in Bangladesh encompasses a wide range of interventions, the implementation lacks strategic grounds. Thus, it is essential to understand the gap and introduce effective sustainable solutions to improve diet quality along with measures to control parasitic infestations in addition to running massive supplementation programs and awareness campaigns.

## Supporting information

**S1 File. Survey questionnaire.**
(PDF)

**S1 Dataset. Dataset of adolescent girls.**
(XLS)

**S2 Dataset. Dataset of pregnant women.**
(XLS)

**S3 Dataset. Dataset of lactating women.**
(XLS)

## Author Contributions

**Conceptualization:** Gulshan Ara, Rafid Hassan, Md. Ahshanul Haque, Anika Bushra Boitchi, Hafizur Rahman.

**Data curation:** Rafid Hassan, Samira Dilruba Ali.

**Formal analysis:** Rafid Hassan.

**Funding acquisition:** Gulshan Ara, Md. Ahshanul Haque, Anika Bushra Boitchi, Hafizur Rahman.

**Investigation:** Gulshan Ara, Md. Ahshanul Haque, Anika Bushra Boitchi, Hafizur Rahman.

**Methodology:** Gulshan Ara, Rafid Hassan, Md. Ahshanul Haque, Anika Bushra Boitchi, Hafizur Rahman.

**Project administration:** Gulshan Ara, Rafid Hassan, Md. Ahshanul Haque, Anika Bushra Boitchi, Samira Dilruba Ali.

**Resources:** Gulshan Ara, Rafid Hassan, Md. Ahshanul Haque, Anika Bushra Boitchi, Hafizur Rahman.

**Software:** Gulshan Ara, Rafid Hassan, Samira Dilruba Ali.

**Supervision:** Gulshan Ara, Rafid Hassan, Md. Ahshanul Haque, Anika Bushra Boitchi, Samira Dilruba Ali.

**Validation:** Gulshan Ara, Rafid Hassan, Md. Ahshanul Haque, Anika Bushra Boitchi, Samira Dilruba Ali.

**Visualization:** Gulshan Ara, Rafid Hassan.

**Writing – original draft:** Gulshan Ara, Rafid Hassan, Anika Bushra Boitchi, Kazi Sudipta Kabir.

**Writing – review & editing:** Gulshan Ara, Rafid Hassan, Md. Ahshanul Haque, Anika Bushra Boitchi, Samira Dilruba Ali, Kazi Sudipta Kabir, Riad Imam Mahmud, Kazal Ahidul Islam, Hafizur Rahman, Zhahirul Islam.

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
