## [Decision Letter · Decision Letter 0]

16 Jan 2024

PONE-D-23-35666Anaemia among adolescent girls, pregnant and lactating women in the south coastal region of Bangladesh: prevalence and risk factorsPLOS ONE

Dear Dr. Ara,

Thank you for submitting your manuscript to PLOS ONE. After careful consideration, we feel that it has merit but does not fully meet PLOS ONE’s publication criteria as it currently stands. Therefore, we invite you to submit a revised version of the manuscript that addresses the points raised during the review process.

We look forward to receiving your revised manuscript.

Kind regards,

Biswajit Pal, M.SC., Ph.D

Academic Editor

PLOS ONE

Journal Requirements:

3.  In the online submission form, you indicated that [The data that support the findings of this study are available on request from the corresponding author. The data are not publicly available due to privacy or ethical restrictions.]. 

Additional Editor Comments:

The study was conducted in Max Nutri WASH program zones but the present title of the research generalizes the issue. Kindly reconsider the title.

Kindly address the issues raised by the reviewers.

Reviewers' comments:

Reviewer's Responses to Questions

**Comments to the Author**

1. Is the manuscript technically sound, and do the data support the conclusions?

Reviewer #1: Yes

Reviewer #2: Yes

2. Has the statistical analysis been performed appropriately and rigorously? 

Reviewer #1: Yes

Reviewer #2: Yes

3. Have the authors made all data underlying the findings in their manuscript fully available?

Reviewer #1: No

Reviewer #2: Yes

4. Is the manuscript presented in an intelligible fashion and written in standard English?

Reviewer #1: Yes

Reviewer #2: Yes

5. Review Comments to the Author

Reviewer #1: The manuscript is technically sound and written in a standard english. It is an important scientific research as anaemia is a serious public health problem among girls and women within reproductive age of the developing countries. The data also supported the conclusion and statistical analysis was done appropriately. There are some restricitions about the data availability. The author must be specified about the data availability statements. Need to revise the keywords and syntax error.

Reviewer #2: Introduction:

• Line 55-56: “Anaemia is determined by a decreasing red blood cell count below the cut-off point set by the World Health Organization”- is a wrong statement. What WHO says is- “Anaemia is a condition in which the number of red blood cells or the haemoglobin concentration within them is lower than normal”. Please use correct clinical terms to define a clinical condition like anaemia.

• Line 76: Reference missing.

• Line 86: Grammatically incorrect.

• Line 87: Reference missing.

• Line 92: Grammatically incorrect.

• What is the justification of conducting the survey in south coastal region of Bangladesh?

• In title, the region is mentioned as the south coastal region of Bangladesh whereas in line 101 the region is mentioned as rural southern region. Which one is correct? Please be consistent.

Methods:

• How is the “"Max Nutri-WASH" initiative connected to the reported study? This is unclear. Please detail this.

• Line 117: desired precision of what? Non-response rate of what? Please mention specifically.

• Line 123: standard or standardized? How did the authors standardize the questionnaire? Please clarify.

• Please add the English and Bangla versions of the questionnaire as supplementary files.

• Line 144-145: Reference missing.

• Ethical considerations, Line 164: “assent (<18 years) from their parents/caregivers”. But assent should be taken from the adolescents and that will be supported by subsequent consent taken from their caregivers. Please check and modify.

• Ethical considerations, Line 166-167: Giving assurance about maintaining confidentiality is important, but maintaining the confidentiality is more important. Was it maintained? Please mention.

• Statistical analysis: Predictors or explanatory variables or independent variables? Please use only one term consistently.

Results:

• Line 183. Five or more groups out of how many food groups? Please be specific and detail the MDD-W cut-off in the relevant section.

• Were all the clusters of same size? If not, were the reported proportions and the estimates weighted as proportional to size? Please explain.

6. PLOS authors have the option to publish the peer review history of their article (what does this mean?). If published, this will include your full peer review and any attached files.

Reviewer #1: No

Reviewer #2: No

---

## [Author Response · Author response to Decision Letter 0]

19 Feb 2024

Editor Comments: In the online submission form, you indicated that [The data that support the findings of this study are available on request from the corresponding author. The data are not publicly available due to privacy or ethical restrictions.]. 

Response to Editor: All relevant data are attached within the manuscript and its Supporting Information files in the revised version.

Additional Editor Comments: The study was conducted in Max Nutri WASH program zones but the present title of the research generalizes the issue. Kindly reconsider the title. Kindly address the issues raised by the reviewers.

Response to Editor: The participants of the study were the beneficiaries of the Max Nutri WASH program and the program was implemented in southern region. Therefore, in the title we mentioned southern region. In the response to the reviewer sections, we addressed all the issues raised by the reviewers.

Review Comments to the Author

Reviewer #1 comments: The manuscript is technically sound and written in a standard english. It is an important scientific research as anaemia is a serious public health problem among girls and women within reproductive age of the developing countries. The data also supported the conclusion and statistical analysis was done appropriately. There are some restrictions about the data availability. The author must be specified about the data availability statements. Need to revise the keywords and syntax error.

Response to Reviewer #1: Thank you for your constructive feedback. In the revised version, all relevant dataset is attached. Additionally, we carefully revise the keywords and address the syntax error.

Reviewer #2 comments: Introduction: Line 55-56: “Anaemia is determined by a decreasing red blood cell count below the cut-off point set by the World Health Organization”- is a wrong statement. What WHO says is- “Anaemia is a condition in which the number of red blood cells or the haemoglobin concentration within them is lower than normal”. Please use correct clinical terms to define a clinical condition like anaemia.

Response to Reviewer #2: In line 55-56: We revised the definition of anaemia according to the reviewer’s suggestion- “This adverse health condition arises when the red blood cells count or the concentration of haemoglobin within them falls below normal.”

Reviewer #2 comments: Line 76: Reference missing.

Response to Reviewer #2: In the revised version, reference is added to this line. 

Reviewer #2 comments: Line 86: Grammatically incorrect.

Response to Reviewer #2: Sorry for this inconsistency. We have corrected the grammatical issue. The revised line 86: In Bangladesh, anaemia affected around half of the adolescent girls (52%) [4], pregnant (50%) and lactating (49%) women.

Reviewer #2 comments: Line 87: Reference missing.

Response to Reviewer #2: In the revised version, reference is added to this line.

Reviewer #2 comments: Line 92: Grammatically incorrect.

Response to Reviewer #2: Sorry for this inconsistency. We have corrected the grammatical issue. The revised line 92: This lower prevalence of iron deficiency anaemia suggests other possible factors that could explain the occurrence of anaemia in Bangladesh.

Reviewer #2 comments: What is the justification of conducting the survey in south coastal region of Bangladesh?

Response to Reviewer #2: The study was conducted in Max Nutri-WASH programme area. The programme is mainly focused southern rural part of Bangladesh. The southern part has both coastal and non-coastal areas. Our study participants were from both coastal and non-coastal areas. Therefore, we revised the title as “southern rural region”. The study was funded by Max-Foundation, they wanted to measure the anaemia situation of the beneficiaries of Max Nutri-WASH programme. Besides, there is a scarcity of recent evidence of anaemia and its determinants among adolescent girls, pregnant, and lactating women in rural community settings of Bangladesh, especially in the southern region. This area, being climate-vulnerable, is more susceptible to micronutrient deficiencies. Therefore, this study was conducted in this region.

Reviewer #2 comments: In title, the region is mentioned as the south coastal region of Bangladesh whereas in line 101 the region is mentioned as rural southern region. Which one is correct? Please be consistent.

Response to Reviewer #2: The study was conducted in Max Nutri-WASH programme area. The programme is mainly focused southern rural part of Bangladesh. The southern part has both coastal and non-coastal areas. Our study participants were from both coastal and non-coastal areas. Therefore, we revised the title as “southern rural region” and it was kept consistent throughout the manuscript.

Reviewer #2 comments: Methods: How is the “"Max Nutri-WASH" initiative connected to the reported study? This is unclear. Please detail this.

Response to Reviewer #2: Reproductive age women and children are more vulnerable to micronutrient malnutrition, especially anemia. Inadequate sanitation increases the probability of parasitic diseases, which generates iron deficiency, and WASH conditions may influence the prevalence of anemia. Max Foundation has been implementing healthy village campaign program in "Max Nutri-WASH" Program areas in 62 Unions of five districts- Patuakhali, Barguna, Khulna, Jessore, and Satkhira on water, sanitation, and hygiene, nutrition, adolescent and women reproductive health. This cross-sectional study was conducted among the beneficiaries who were enlisted to the "Max Nutri-WASH" program in three southern districts (Khulna, Patuakhali, and Satkhira) of Bangladesh.

Reviewer #2 comments: Line 117: desired precision of what? Non-response rate of what? Please mention specifically.

Response to Reviewer #2: Desired precision or margin of error was 6%. Non-response was considered to 10% which refers to participants who were not willing to provide blood samples for haemoglobin measurement or did not continue the full interview session.

Reviewer #2 comments: Line 123: standard or standardized? How did the authors standardize the questionnaire? Please clarify.

Response to Reviewer #2: We apologies for this inconsistency. It would be standard not standardized. The line was revised as: A standard structured questionnaire following the questionnaire of the National Micronutrient Survey of Bangladesh, Bangladesh Demographic Health Survey was formulated to collect data from the study participants.

Reviewer #2 comments: Please add the English and Bangla versions of the questionnaire as supplementary files.

Response to Reviewer #2: The English and Bangla versions of the questionnaire are added as a supplementary file.

Reviewer #2 comments: Line 144-145: Reference missing.

Response to Reviewer #2: In the revised version, reference is added to this line.

Reviewer #2 comments: Ethical considerations, Line 164: “assent (<18 years) from their parents/caregivers”. But assent should be taken from the adolescents and that will be supported by subsequent consent taken from their caregivers. Please check and modify.

Response to Reviewer #2: We modified the Ethical considerations in the revised version. It was written as: The institutional review board (IRB) of icddr,b approved the study (protocol # PR-21124). Prior to the interview, informed written consent was taken in the local language from the study participants (>18 years). Assent was taken from the adolescent girls whose age was below 18 years and subsequently, consent was obtained from their parents and caregivers. Participants who were illiterate provided consent by thumb impression. The research team described to the participants about background and objectives of the study, the voluntary nature of participation, and the future use of data. They were given assurance that confidentiality would be maintained for all the gathered information. None other than the investigators of this research, the Ethical Review Committee of icddr,b, and any law-enforcing agency in the event of necessity would have access to the information.

Reviewer #2 comments: Ethical considerations, Line 166-167: Giving assurance about maintaining confidentiality is important, but maintaining the confidentiality is more important. Was it maintained? Please mention.

Response to Reviewer #2: We assured that the privacy, anonymity and confidentiality of data/information identifying the study subjects was strictly maintained. We kept all the collected information and results of the clinical tests performed on confidential, under lock and key. None other than the investigators of this research, the Ethical Review Committee of icddr,b and any law-enforcing agency in the event of necessity would have an access to the information.

Reviewer #2 comments: Statistical analysis: Predictors or explanatory variables or independent variables? Please use only one term consistently.

Response to Reviewer #2: In the revised version, independent variables is consistently used. 

Reviewer #2 comments: Results: Line 183. Five or more groups out of how many food groups? Please be specific and detail the MDD-W cut-off in the relevant section.

Response to Reviewer #2: A diet was considered diversified if anyone consumed at least five of the ten food groups. It was elaborated in the methods section. The revised lines: The Minimum Dietary Diversity-Women (MDD-W) indicator was used to assess their dietary diversity, which was developed based on a preceding 24-hour dietary recall from a list of ten selected food groups, including grains/roots/tubers, pulses/legumes, nuts/seeds, dairy, eggs, meat/poultry/fish, dark green leafy vegetables, other vitamin A-rich fruits and vegetables, other fruit, and other vegetables. A diet was considered diversified if anyone consumed at least five of the ten food groups.

Reviewer #2 comments: Were all the clusters of same size? If not, were the reported proportions and the estimates weighted as proportional to size? Please explain.

Response to Reviewer #2: Yes, all the clusters were same in size. Here, the cluster was union and equal number of participants from each target group was chosen using a systematic sampling technique.

---

## [Decision Letter · Decision Letter 1]

13 Jun 2024

Anaemia among adolescent girls, pregnant and lactating women in the southern rural region of Bangladesh: prevalence and risk factors

PONE-D-23-35666R1

Dear Dr. Ara,

We’re pleased to inform you that your manuscript has been judged scientifically suitable for publication and will be formally accepted for publication once it meets all outstanding technical requirements.

Kind regards,

Biswajit Pal, M.SC., Ph.D

Academic Editor

PLOS ONE

Additional Editor Comments (optional):

Reviewers' comments:

Reviewer's Responses to Questions

**Comments to the Author**

1. If the authors have adequately addressed your comments raised in a previous round of review and you feel that this manuscript is now acceptable for publication, you may indicate that here to bypass the “Comments to the Author” section, enter your conflict of interest statement in the “Confidential to Editor” section, and submit your "Accept" recommendation.

Reviewer #1: All comments have been addressed

2. Is the manuscript technically sound, and do the data support the conclusions?

Reviewer #1: Yes

3. Has the statistical analysis been performed appropriately and rigorously? 

Reviewer #1: Yes

4. Have the authors made all data underlying the findings in their manuscript fully available?

Reviewer #1: Yes

5. Is the manuscript presented in an intelligible fashion and written in standard English?

Reviewer #1: Yes

6. Review Comments to the Author

Reviewer #1: Authors have adequately addressed the comments raised in a previous round of review and this manuscript is now acceptable for publication.

7. PLOS authors have the option to publish the peer review history of their article (what does this mean?). If published, this will include your full peer review and any attached files.

Reviewer #1: No

---

## [Editor Report · Acceptance letter]

1 Jul 2024

PONE-D-23-35666R1 

PLOS ONE

Dear Dr. Ara, 

I'm pleased to inform you that your manuscript has been deemed suitable for publication in PLOS ONE. Congratulations! Your manuscript is now being handed over to our production team.

Kind regards, 

on behalf of

Dr. Biswajit Pal 

Academic Editor

PLOS ONE